# easyFulcrum: An R package to process and analyze ecological sampling data generated using the Fulcrum mobile application

**Matteo Di Bernardo**⦿◯, **Timothy A. Crombie**◯, **Daniel E. Cook**, **Erik C. Andersen**⦿*

Department of Molecular Biosciences, Northwestern University, Evanston, IL, United States of America

◯ These authors contributed equally to this work.
* erik.andersen@northwestern.edu

**Data Availability Statement:** All source code and example data for the easyFulcrum package are available in a GitHub repository (https://github.com/AndersenLab/easyFulcrum). A vignette of the

## Abstract

Large-scale ecological sampling can be difficult and costly, especially for organisms that are too small to be easily identified in a natural environment by eye. Typically, these microscopic floral and fauna are sampled by collecting substrates from nature and then separating organisms from substrates in the laboratory. In many cases, diverse organisms can be identified to the species-level using molecular barcodes. To facilitate large-scale ecological sampling of microscopic organisms, we used a geographic data-collection platform for mobile devices called Fulcrum that streamlines the organization of geospatial sampling data, substrate photographs, and environmental data at natural sampling sites. These sampling data are then linked to organism isolation data from the laboratory. Here, we describe the easyFulcrum R package, which can be used to clean, process, and visualize ecological field sampling and isolation data exported from the Fulcrum mobile application. We developed this package for wild nematode sampling, but it can be used with other organisms. The advantages of using Fulcrum combined with easyFulcrum are (1) the elimination of transcription errors by replacing manual data entry and/or spreadsheets with a mobile application, (2) the ability to clean, process, and visualize sampling data using a standardized set of functions in the R software environment, and (3) the ability to join disparate data to each other, including environmental data from the field and the molecularly defined identities of individual specimens isolated from samples.

## Introduction

Ecological studies of small, difficult to identify species are challenging because researchers are effectively unable to see these species at sampling sites in nature. The ecology of most of these species can only be studied by sampling substrates from the wild and then separating the species of interest from substrates in the laboratory. The effort and time required to identify species in the laboratory can make it difficult to accurately connect the identified specimens back to the ecological data recorded at the site of collection and can hinder studies of natural populations. Variations of this laborious sampling strategy are used to study the ecology of many

easyFulcrum workflow with example data in the raw csv and jpg format is available online (http://andersenlab.org/easyfulcrum/articles/easyFulcrum.html). The easyFulcrum R package is open source, and we encourage users to open issues or comment on the GitHub repository.

**Funding:** This project and equipment were funded using start-up funds from the Molecular Biosciences department and Weinberg College of Arts and Sciences at Northwestern University. E.C. A. was funded by an NSF Division of Integrated Organismal Systems CAREER award (1751035). The funders had no role in study design, data collection and analysis, decision to publish, or preparation of the manuscript.

**Competing interests:** The authors have declared that no competing interests exist.

prominent model organisms, including *Caenorhabditis elegans*, *Drosophila melanogaster*, and *Saccharomyces cerevisiae*, and likely contribute to the comparative sparsity of ecological data for these species relative to nearly every other aspect of their biology [1–3]. Here, we address some of the difficulties associated with ecological sampling of small, difficult to identify species by leveraging mobile data collection platforms, cloud-based databases, and the R software environment.

Fulcrum is a customizable, data-collection platform compatible with Apple iOS and Google Android devices that allows users to collect rich, location-based data (https://www.fulcrumapp.com). It is a commercial application but can be used under a no-cost educational agreement. To facilitate large-scale ecological surveys of nematodes that are difficult to identify in the field, we developed two custom Fulcrum applications. The "Nematode field sampling" application allows the user to organize various ecological data types associated with the substrates sampled in the field, such as environmental parameters and substrate characteristics, using their mobile device. The "Nematode isolation" application helps organize data associated with the specimens isolated from samples after they have been brought into the laboratory [4]. Importantly, these applications are easily extensible to other organisms because Fulcrum uses a powerful GUI to help users customize data-collection applications even when they have no coding or database administration knowledge. This utility makes it easy to use Fulcrum's robust, cloud-based database for sampling nearly any species from nature. Two key advantages of our approach are improvements in sampling efficiency and data organization, which translate into more samples collected by each researcher and greater accuracy when linking isolated specimens back to the ecological data at the sampling site.

Here, we describe the easyFulcrum R package, which contains a suite of functions designed to process and analyze data exported from Fulcrum and join these data with genotype information if organisms are identified using molecular barcodes. The easyFulcrum package uses standard R packages to rapidly read data into R, flag anomalies, join diverse data sources, and reformat data in a tidy format [5]. The package also includes functions to review these data in markdown reports, providing summary statistics, notes on potential anomalies, and interactive collection maps for discrete sampling projects. When combined, the Fulcrum data-collection platform and easyFulcrum R package are a powerful tool for collecting and processing ecological data in a simple, standardized format that can be employed by researchers with limited backgrounds in computer and data science.

## Methods

### Fulcrum and application customization

The Fulcrum data collection application for Apple iOS or Google Android devices can be downloaded online (https://www.fulcrumapp.com). Fulcrum uses a powerful GUI to allow users to create their own mobile applications. We created two different applications: Nematode field sampling and Nematode isolation. However, users can create their own applications for field sampling and isolation. When creating these applications, they can follow our Fulcrum templates and save their applications with a unique identifier followed by either "field sampling" or "isolation". The template for the Nematode field sampling application is here: [https://www.fulcrumapp.com/apps/nematode-field-sampling] and the Nematode isolation template is here: [https://www.fulcrumapp.com/apps/nematode-isolation]. We provide the minimum requirements for customized field sampling and isolation applications to work with easyFulcrum (Table 1). For easyFulcrum compatibility, custom applications must use the same data names for the required fields. When creating custom applications, the isolation application can be linked to the field sampling application by setting the "Linked App" option in the

**Table 1. Minimum field requirements for customized field sampling and isolation applications.**

| field sampling application | | Isolation application | |
|---|---|---|---|
| **Fields** | **data names** | **Fields** | **data names** |
| Sample photo | sample_photo | C-Label | c_label |
| Gridsect | gridsect | Photos | photos |
| Substrate Temperature (C) | substrate_temperature | S-labeled Plates | s_labeled_plates |
| Ambient Humidity (%) | ambient_humidity | Date | date |
| Ambient Temperature (C) | ambient_temperature_c | Time | time |
| C-Label | c_label | | |
| Date | date | | |
| Time | time | | |

C-label field of the isolation application to the name of your field sampling application. If the Worms on Sample field is not used in the custom isolation application, the "Visibility" and "Requirement" rules in the S-labeled Plates field must be removed. Other than these requirements, any fields can be added to the customized applications. The Fulcrum GUI will help guide application creation and/or editing.

## Field sampling with Fulcrum

Before going into the field, unique collection labels (C-labels) are generated as scannable QR codes to distinguish one collection from another and attached to plastic collection bags. If no QR codes are used, the labels can be entered manually, but manual entry can lead to analysis errors later and is not recommended. In the field, a user will open the Fulcrum "Nematode field sampling" application and use the device camera to scan a C-label QR code from a collection bag to initiate a new collection record. The user then enters data associated with the sample into the fields of the collection record, including various environmental parameter values and photographic evidence of the sample at the site. This process is then repeated until the desired number of samples are collected. Once the samples are collected, they are brought to the laboratory and enter the specimen isolation workflow. The Nematode field sampling application expedites data recording by automatically recording the GPS location when the user initiates a new collection record. If a new collection record is created away from the sample collection site, the correct GPS location for the collection site can be extracted from the metadata associated with the photo of the sample. Importantly, all the data entered into the collection records are kept locally on the mobile device, whether the device is connected to cellular service or not, and it can be synchronized to the cloud at a later time when service is restored. This feature is useful for field sampling in remote locations. Moreover, with Fulcrum, teams of samplers can work independently at different locations using separate mobile devices, and all the collection records can be synchronized to the cloud database.

## Specimen isolation with Fulcrum

The samples collected from the field are processed in a field station or laboratory using the Fulcrum Nematode isolation application. Here, the user initiates an isolation record for a particular collection by scanning the C-label QR code attached to the collection bag with the mobile device camera. This step ensures that the isolation record is linked to the correct collection record in the Fulcrum database. The user then enters information about the condition of the sample and the estimated number of organisms that are present on the sample. The user then transfers organisms to isolation containers that are pre-barcoded with unique isolation labels

(S-labels). An isolation label is scanned into the isolation record for each isolated organism so that all specimens isolated from the sample can be traced back to the original collection record. After isolation, the specimens can be identified by analysis of morphology or by sequence similarity using molecular barcodes.

## Specimen identification

We built easyFulcrum to read specimen identification data from a Google sheet, which we refer to as the genotyping sheet. We chose to use a Google sheet for these data rather than Fulcrum because we identify nematodes by sequence similarity using molecular barcodes and found it easier to track the genotyping data for S-labels in an online spreadsheet. We provide two templates for the genotyping sheet, users that wish to use a simple, generalizable genotyping sheet can find the general template here https://docs.google.com/spreadsheets/d/1Soo7R7srnmBu29Z_ZbYweTMg_IdO1N5sWr6cr0NpioY/edit#gid=1595072024. The nematode specific genotyping template can be found here https://docs.google.com/spreadsheets/d/1pvdpujuJDTYaMwjjDlivjAnO0MrQuDguU2kRLs9eemQ/edit#gid=0. To use either of the genotyping sheet templates, right click the "genotyping template" tab on the lower left and select "Copy to new spreadsheet" then select "Open spreadsheet" to set up a new genotyping sheet for your collection project. Next, the S-labels for a collection project are exported from Fulcrum and pasted into the genotyping sheet in the "s_label" column, as described below (see Methods, Fulcrum data export). Once the genotyping sheet has the S-labels entered, the genotyping data are manually entered for each S-label. We include a description of each variable in the general genotyping sheet and how to enter results in Table 2. Similar instructions for the nematode specific genotyping sheet are included in the "column description" tab of the nematode-specific genotyping template. We emphasize that users should generate specimen sequence data and perform alignments to sequence databases outside of easyFulcrum. Then, these species identification labels should be entered into the genotyping sheet (*e.g.*, enter the genus and species name with the best alignment score to the specimen in the species_id column. Users that want to identify nematodes can use the following primer sets to amplify molecular barcodes. The small subunit (SSU) primer set (RHAB1350F: `TACAATGGAAGGCA GCAGGC` and RHAB1868R: `CCTCTGACTTTCGTTCTTGATTAA`) amplifies a 500-bp fragment of the 18S rDNA gene in Rhabditid nematodes [6]. The SSU primer set can be used to place nematodes within Rhabditid groups and check the quality of lysis material since most nematode species will amplify with this primer set. The internal transcribed spacer (ITS2) primer set (oECA1687: `CTGCGTTACTTACCACGAATTGCARAC` and oECA202: `GCGGTATTTGCTACT ACCAYYAMGATCTGC`) amplifies a 2,000-bp fragment of the ITS2 region between the 5.8S and 28S rDNA genes and can be used to identify nematodes to the species level within the *Caenorhabditis* genus [7]. The details of sample lysis, SSU and ITS2 PCR conditions, Sanger

**Table 2. Description of the general genotyping sheet variables.**

| Variable name | Description |
| --- | --- |
| project_id | Enter the project name. The project name should match the project name assigned to the S-label in Fulcrum. |
| s_label | Enter the S-label name. Each row should be a unique S-label. Use Fulcrum data export tools to obtain S-labels and paste them into the sheet (see Methods, Fulcrum data export). |
| species_id | Enter the species name for the top alignment score for the S-label. For species identification, the sequence to amplify and align is up to the user. |
| possible_new_sp | Enter a 1 if the alignment indicates the S-label may be a new species. Leave blank if not. |
| strain_name | Enter a unique name for the isolated strain. Leave blank if no name is to be given. |

sequencing of amplification products, and alignment strategies for nematodes have been documented previously [4, 8] Furthermore, detailed instructions for how to use the nematode-specific genotyping sheet are included in the easyFulcrum package vignette http://andersenlab.org/easyfulcrum/articles/easyFulcrum.html. For non-nematode users, the choice of molecular barcodes will depend on the taxa that they want to identify. Although the construction and testing of molecular barcodes is beyond the scope of this paper, many excellent resources are available for guidance [9].

## Fulcrum data export

Before processing collection data using easyFulcrum, the raw Fulcrum data must be exported from the Fulcrum database using the Fulcrum website's data export tool. We recommend exporting with the following settings; select the checkboxes for the desired project, include photos, include GPS data, field sampling, and isolation. The field sampling and isolation data should be exported from the Fulcrum database in comma-separated value (csv) format and named as follows when exporting from the nematode field sampling and nematode isolation applications.

- nematode_field_sampling_sample_photo.csv

- nematode_field_sampling.csv

- nematode_isolation_photos.csv

- nematode_isolation_s_labeled_plates.csv

- nematode_isolation.csv

If customized Fulcrum applications are used, the [nematode] prefix will be replaced with [your prefix] in the exported csv files. easyFulcum will use any prefix but custom applications must be named [your prefix] field sampling and [your prefix] isolation. The photos are exported as jpg files named with unique alpha-numeric record labels from the Fulcrum database. These files must then be moved to the correct location in the project directory structure to be processed with easyFulcrum, as described below. The easyFulcrum function *makeDirectoryStructure()* will create the required project directory structure for the user. The csv files are moved to *[your project directory]/data/raw/fulcrum*, and the jpg files are moved to *[your project directory]/data/raw/fulcrum/photos*.

## easyFulcrum installation

The easyFulcrum package can be installed easily using the following command in R: devtools::install_github("AndersenLab/easyfulcrum"). Package installation requires R v3.5.0 or later and devtools v2.4.1 or later. For OS X users, easyFulcrum's *procPhotos()* function requires users to install XQuartz outside of R; XQuartz software can be found here: (https://www.xquartz.org). For novice R users, we suggest one of many helpful guides on R programming that includes detailed instructions for installing R and RStudio (https://rstudio-education.github.io/hopr/starting.html) [10].

## Results

### easyFulcrum overview

The easyFulcrum R package is designed to simplify and standardize the processing of ecological field sampling data generated using the Fulcrum mobile application. easyFulcrum contains

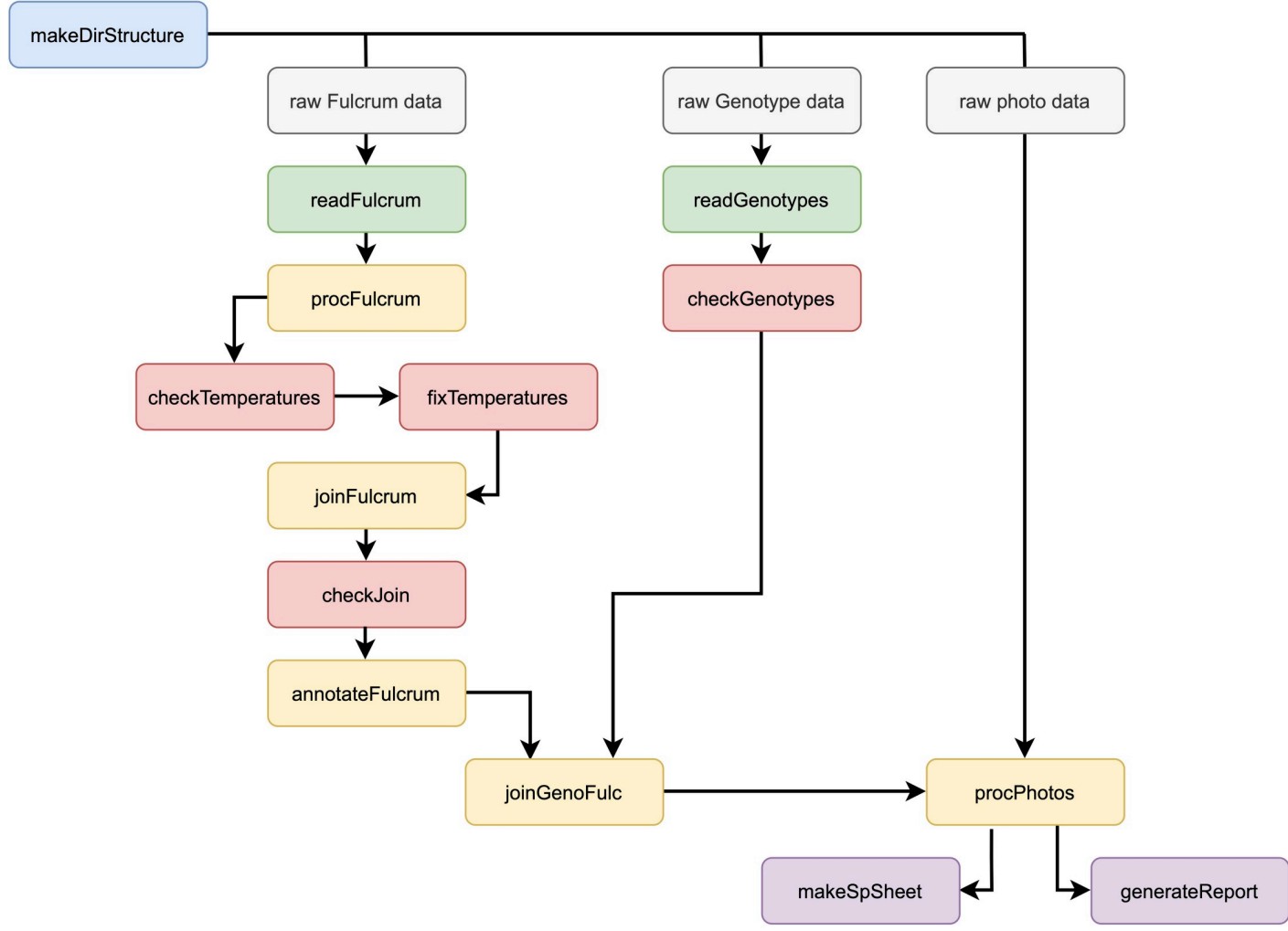

**Fig 1. easyFulcrum workflow.** The suggested workflow for processing a sampling project begins with making a project directory using the *makeDirStructure()* function (blue). The raw Fulcrum and photo data are exported from the Fulcrum database and loaded into the directory structure manually (grey). The raw genotype data is sourced directly from a Google sheet (grey). The workflow for read functions (green), processing functions (yellow), checking functions (red), and summarization functions (purple) is shown.

13 main functions to read, process, check, join, and summarize collection data for a particular sampling project (Fig 1). This software package reads raw data and image files exported from the Fulcrum database and genotype data for isolated organisms sourced from a Google sheet (see Methods, Specimen identification). Below, we describe the workflow to process a typical sampling project using easyFulcrum.

## Reading, processing, and joining Fulcrum data

easyFulcrum uses a defined directory structure to read raw data files exported from Fulcrum into R and to write processed output data. We included the *makeDirStructure()* function to create the required directory structure at a location specified by the user (see Methods, Fulcrum data export, Fig 2). After files are loaded into the directory structure for a collection project, the *readFulcrum()* function reads the raw Fulcrum data files into R as named dataframes and adds them to a single list object.

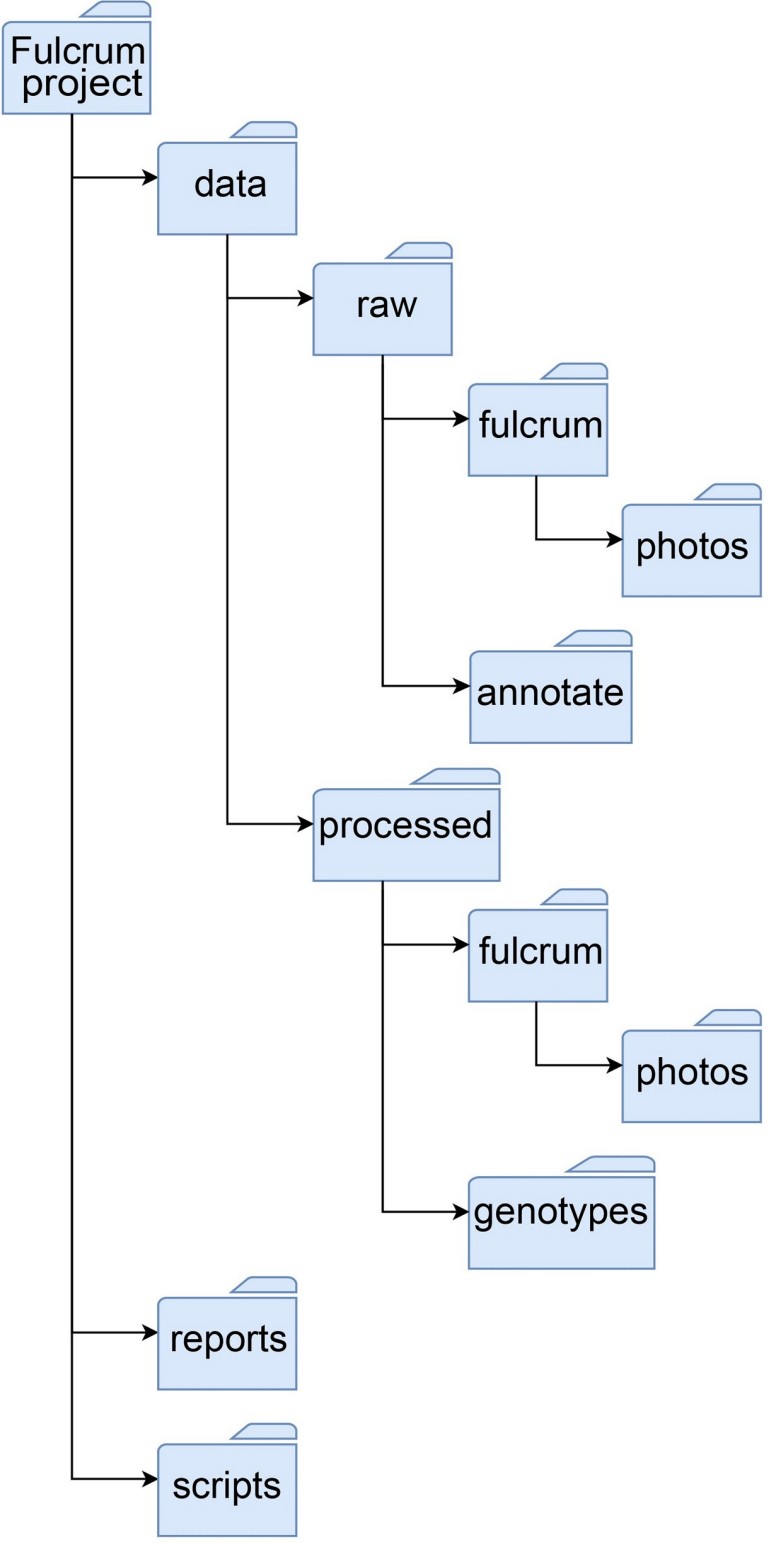

**Fig 2. easyFulcrum directory structure.** The *makeDirStructure()* function will produce the directory structure shown above. In this directory structure, the *data* folder represents the main subfolder containing both raw and processed Fulcrum data. The *data/raw/fulcrum* subfolder contains the five raw comma-separated value csv files exported from the Fulcrum website and the *data/raw/fulcrum/photos* subfolder contains the exported photos in jpg format (see Methods, Fulcrum data export) files exported from Fulcrum. The *data/raw/annotate* subfolder can be used to hold

spatial annotation files that are used by the *annotateFulcrum()* function to relate sampling sites to known geographical features. The *data/processed/fulcrum* subfolder holds the processed collection data and photos exported by easyFulcrum functions. The *reports* directory is the default location for saving collection reports generated with the *generateReport()* function which is described in more detail in the text.

Once the data are read into R, the *procFulcrum()* function can process each of the five data-frames independently to standardize temperature, time, altitude, and location data into defined formats. Furthermore, this function generates 'flag' variables that identify anomalies in the data based on unusual temperature values and misformatted or missing collection labels. By default, *procFulcrum()* converts all temperature values above 40 from Fahrenheit to Celsius, which may not be appropriate for some collection projects, and flags these records. The *check-Temperatures()* function can be used to display flagged temperature values so that the user can decide if the conversion was inappropriate. This function also returns values that are flagged if the temperature probe has recorded identical values across many sequential collections, sug-gesting that mistakes could have occurred during temperature data collection. The *fixTem-peratures()* function allows the user to revert improperly converted temperatures back to the original value after review and to eliminate any erroneous temperature values. Given that tem-perature unit errors are quite frequent, we have provided the easy to use *fixTemperature()* function such that even beginner R users can use the software to correct temperature values. After reverting all values that were improperly converted or eliminating erroneous tempera-ture values, the user can run *checkTemperatures()* again to determine if the flagged values were corrected.

After processing the Fulcrum data and addressing temperature anomalies, the *joinFulcrum ()* function is used to join the processed Fulcrum dataframes together. The function first joins the field_sampling dataframe to the isolation dataframe using unique alpha-numeric collection and isolation record identifiers exported from the Fulcrum database. Following this step, *join-Fulcrum()* then selects the best photo for each collection record. If multiple photos exist for a single collection record, the best photo is chosen based on an estimate of GPS accuracy, which is extracted from the photo metadata in the field_sampling_sample_photo dataframe. Most modern mobile phones have a built-in GPS receiver that stores location information in the photo metadata when a picture is taken. The "exif_gps_dop" variable is a measure related to the GPS degree of precision that is exported from Fulcrum in units of meters. We select the photo with the smallest value corresponding to the highest GPS degree of precision. Once the best photos are selected, the unique photo record identifiers and GPS locations from the best photos are joined to the previous joined dataframe. *joinFulcrum()* then adds two location vari-ables to the joined dataframe, one for the best photo GPS locations, and another for the GPS locations recorded at the time the collection records were generated. We recommend users prioritize the best photo GPS locations when available because we have found that they are typ-ically closer to the actual sampling sites than the GPS locations recorded at the time the collec-tion records were generated using Fulcrum. The S-labels are then added when the processed isolation_s_labeled_plates dataframe is joined to the previous dataframe on the basis of the unique isolation records. Finally, the *joinFulcrum()* function adds flags for extreme tempera-tures and altitude values, duplicated or missing C-labels or S-labels, and variables indicating whether the final collection location should be taken from the photo metadata or from the location at the time the collection record was generated. If needed, the *joinFulcrum()* function can handle the special case where isolation data are not generated and only the field_sampling and field_sampling_sample_photo dataframes exist. The *checkJoin()* function will display flags and indicate which of the input Fulcrum data files had errors. Because the R interface does not

provide easy cell-by-cell manipulations once data are entered into the R workspace, the best method to fix errors is to alter flagged errors in R using the functions provided in easyFulcrum. Raw collection or isolation data should never be altered.

Finally, the optional *annotateFulcrum()* function maps collection locations to geographic features that the user can input based on the processed longitude and latitude of the sample. This function allows the user to input bounding boxes or polygons for known geographic features such as islands, trails, parks, and mountains. The *annotateFulcrum()* function acts as a wrapper for the *sp*::*over* function and will determine if a collection is located within any of the supplied geographic features and add these location descriptions to the dataframe [11]. Users that are unfamiliar with geospatial data can create geojson polygon points via a simple online mapping service [https://boundingbox.klokantech.com], which can be supplied to *annotateFulcrum()* following the easyFulcrum vignette.

## Reading, processing, and joining genotyping results

After the processing and joining of Fulcrum data from the field sampling and laboratory isolation applications, easyFulcrum joins these data with genotyping data for the specimens isolated from the field samples. The genotyping data, often derived from Sanger sequence of isolated organisms, is pulled from a standardized Google Sheet format and filled with collection-specific data generated by the user. We refer to this data source as the genotyping sheet and describe in detail how to enter genotyping data in Methods (see Methods, Specimen identification). The *readGenotypes()* function performs the import of the genotyping sheet by using the *googlesheets4* R package developed for this purpose [12]. Once the genotyping data are read into R, the *checkGenotypes()* function searches the genotyping data, flagging isolations with missing, improper, and/or duplicated S-labels, and performs other checks, including if the species description, strain name, proliferation label, or ITS2 genotype are missing. The *checkGenotypes()* function also reads the joined Fulcrum data to check if disparities exist between the S-labels in the genotyping and Fulcrum data. Again, given limited R functionalities in cell-by-cell manipulations, we suggest for the user to make edits to data programmatically using functions provided in easyFulcrum. Following these checks, the *joinGenoFulc()* function joins the Fulcrum and genotyping data by their shared S-labels. This function can also save an RDS file of the processed genotyping data in */data/processed/genotypes* for fast data import in the future.

## Reading, processing, and joining collection images

Once the genotyping and Fulcrum data are joined, the *procPhotos()* function can be used to copy, rename, and generate thumbnails of raw image files in *data/raw/fulcrum/photos*, so that they can be viewed more easily in interactive reports or online. The *procPhotos()* function renames the photos to the C-label names rather than the unique record identifiers names exported from Fulcrum. The renamed, full-size images are saved in the */data/processed/fulcrum/photos* subdirectory and the renamed thumbnails are saved in the */data/processed/fulcrum/photos/thumbnails* subdirectory. This function also allows for collection photos to be processed to fit criteria for specific community-wide data repositories, like the *C. elegans* Natural Diversity Resource [13]. The final output to the R environment is a single data frame identical to the output from *joinGenoFulc()* but with variables added to describe the original and processed file paths for all sample photos.

## Generating summary and output files

The *generateReport()* function can be used to generate a summary report for a collection project. This function passes collection data to an example R markdown file *sampleReport.Rmd*,

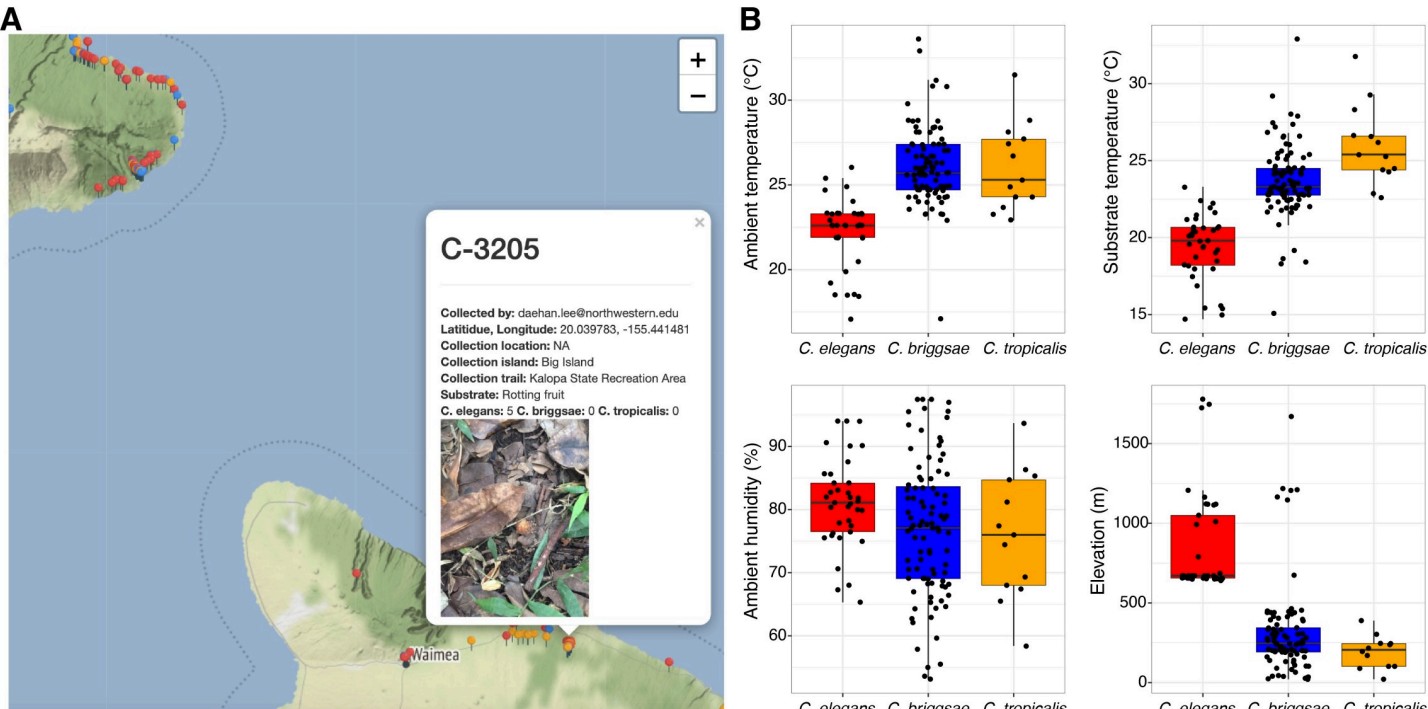

**Fig 3. Sample report attributes.** The *generateReport()* function will render an HTML file with interactive maps of the collections, summary plots of the collection data, and other sections summarizing a collection project. (A) A screenshot of the interactive map with a pop-up caption for a specific collection label C-3205 is shown. The substrate photo is included in the pop-up. The pins on the map represent distinct collection sites and are colored to indicate the presence of nematodes on the sample (red), no nematodes on the sample (blue), only nematode tracks on the sample (orange). This map contains information from OpenStreetMap and OpenStreetMap Foundation, which is made available under the Open Database License. (B) Environmental parameter values, including ambient temperature, substrate temperature, ambient humidity, and elevation for collection sites where *Caenorhabditis* nematodes were found are shown. Tukey box plots are plotted by species (colors) for each environmental parameter. The processed data used to generate the HTML report shown in this figure and a full version of the report are available (S1 Data and S1 File).

which is contained in the package. The *generateReport()* function will write the example *sampleReport.Rmd* to the */scripts* subfolder and knit a report in HTML format to the */reports* subfolder, which can be reviewed in any web browser (Fig 3). The *sampleReport.Rmd* file is provided to the user as a project summary template and can be edited as the user requires. The *sampleReport.Rmd* function includes code to summarize metadata relating to the collection and isolation, such as who conducted the respective processes and on what dates steps were completed. The markdown also provides summary tables of data relating to the collections and isolations. Additionally, we included functions to generate interactive maps of sampling sites (Fig 3A) and box plots of environmental parameters recorded at sampling sites, such as substrate temperature, ambient temperature, humidity, and elevation using the *ggplot2* and *leaflet* R packages (Fig 3B and S1 Data and S1 File) [14, 15]. The workflow vignette can be used to generate a project summary HTML report using the example data included in the package.

## Conclusions

The easyFulcrum R package offers an organized workflow for processing ecological sampling data generated using the Fulcrum mobile application. The package provides simple and efficient functions to clean, process, and visualize ecological field sampling and isolation data collected using custom Fulcrum applications. It also provides functions to join these data with genotype information if organisms isolated from the field are identified using molecular barcodes. Together, the Fulcrum mobile application and easyFulcrum R package allow

researchers to easily implement mobile data-collection, cloud-based databases, and standardized data analysis tools to improve ecological sampling accuracy and efficiency, while simultaneously enabling reproducible analysis and downstream integration with other R packages.

## Supporting information

**S1 Data. The processed data used to generate Fig 3.**
(CSV)

**S1 File. The full HTML report shown in Fig 3.**
(HTML)

## Acknowledgments

We would like to thank members of the Andersen laboratory past and present for their helpful suggestions and feedback developing easyFulcrum.

## Author Contributions

**Conceptualization:** Matteo Di Bernardo, Timothy A. Crombie, Daniel E. Cook, Erik C. Andersen.

**Data curation:** Matteo Di Bernardo, Timothy A. Crombie, Erik C. Andersen.

**Formal analysis:** Matteo Di Bernardo, Timothy A. Crombie.

**Funding acquisition:** Erik C. Andersen.

**Investigation:** Matteo Di Bernardo, Timothy A. Crombie, Erik C. Andersen.

**Methodology:** Matteo Di Bernardo, Timothy A. Crombie, Daniel E. Cook, Erik C. Andersen.

**Project administration:** Erik C. Andersen.

**Resources:** Matteo Di Bernardo, Timothy A. Crombie, Erik C. Andersen.

**Software:** Matteo Di Bernardo, Timothy A. Crombie, Daniel E. Cook.

**Supervision:** Erik C. Andersen.

**Validation:** Matteo Di Bernardo, Timothy A. Crombie.

**Visualization:** Matteo Di Bernardo, Timothy A. Crombie, Daniel E. Cook.

**Writing – original draft:** Matteo Di Bernardo, Timothy A. Crombie, Erik C. Andersen.

**Writing – review & editing:** Matteo Di Bernardo, Timothy A. Crombie, Erik C. Andersen.

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
