## [Decision Letter · Decision Letter 0]

4 Aug 2021

PONE-D-21-18748

easyFulcrum: An R package to process and analyze ecological sampling data generated using the Fulcrum mobile application

PLOS ONE

Dear Dr. Andersen,

Thank you for submitting your manuscript to PLOS ONE. After careful consideration, we feel that it has merit but does not fully meet PLOS ONE’s publication criteria as it currently stands. Therefore, we invite you to submit a revised version of the manuscript that addresses the points raised during the review process.

We look forward to receiving your revised manuscript.

Kind regards,

Syed Ahmad Chan Bukhari

Academic Editor

PLOS ONE

Journal Requirements:

2. Your abstract cannot contain citations. Please only include citations in the body text of the manuscript, and ensure that they remain in ascending numerical order on first mention.

3. We note that Figure 1 in your submission contain copyrighted images. All PLOS content is published under the Creative Commons Attribution License (CC BY 4.0), which means that the manuscript, images, and Supporting Information files will be freely available online, and any third party is permitted to access, download, copy, distribute, and use these materials in any way, even commercially, with proper attribution. For more information, see our copyright guidelines: http://journals.plos.org/plosone/s/licenses-and-copyright.

2. If you are unable to obtain permission from the original copyright holder to publish these figures under the CC BY 4.0 license or if the copyright holder’s requirements are incompatible with the CC BY 4.0 license, please either i) remove the figure or ii) supply a replacement figure that complies with the CC BY 4.0 license. Please check copyright information on all replacement figures and update the figure caption with source information. If applicable, please specify in the figure caption text when a figure is similar but not identical to the original image and is therefore for illustrative purposes only

4. We note that Figure 1 in your submission contain map images which may be copyrighted. All PLOS content is published under the Creative Commons Attribution License (CC BY 4.0), which means that the manuscript, images, and Supporting Information files will be freely available online, and any third party is permitted to access, download, copy, distribute, and use these materials in any way, even commercially, with proper attribution. For these reasons, we cannot publish previously copyrighted maps or satellite images created using proprietary data, such as Google software (Google Maps, Street View, and Earth). For more information, see our copyright guidelines: http://journals.plos.org/plosone/s/licenses-and-copyright.

Reviewers' comments:

Reviewer's Responses to Questions

**Comments to the Author**

1. Is the manuscript technically sound, and do the data support the conclusions?

Reviewer #1: Yes

Reviewer #2: Yes

2. Has the statistical analysis been performed appropriately and rigorously? 

Reviewer #1: Yes

Reviewer #2: N/A

3. Have the authors made all data underlying the findings in their manuscript fully available?

Reviewer #1: No

Reviewer #2: No

4. Is the manuscript presented in an intelligible fashion and written in standard English?

Reviewer #1: Yes

Reviewer #2: No

5. Review Comments to the Author

Reviewer #1: The paper by Bernardo et al. explains the data-collection platform Fulcrum and easyFulcrum applications to organize ecological sample data collection. These applications are easy to use using mobile devices, and helps with identification of microscopic organisms based on their location, photographs and genotyping. This paper focuses on easyFulcrum R package which can analyze exported data from Fulcrum and process the data by combining geographical and sequencing data. These applications are designed for nematodes but can be used for any other organism.

Review Points:

1. In line 53 authors explain about Fulcrum but no reference paper is added.

2. In line 126, authors mention Genotyping template (google sheet) to enter data for isolated samples. Author explain well how to duplicate and use the template but fail to explain columns like ssu_pcr_date, its2_pcr_date etc. Also, there is no mention of the column name to input sanger sequencing data.

3. Authors need to explain how to input genotyping data and blast results if any new species is identified.

4. In line 176, author is suggesting to see methods. It will be helpful if authors mention which subheading in the methods to look for.

5. Authors should mention that Figure 1 is also published on the following website:

6. Figure 3A image quality is not good. Labels in the image are not legible.

7. For Figure 3, authors can include the “genotyping template” made for C.elegans, C.briggsae and C. tropicalis.

Reviewer #2: The manuscript sounds technically poor, I have following concerns should be addressed before any decision.

1. There are some typos and grammatical errors in the manuscript. It is strongly suggested that the whole work to be carefully checked by someone has expertise in technical English writing.

2. Key contribution and novelty has not been detailed in manuscript. Please include it in the introduction section

3. What are the limitations of the related works

4. Are there any limitations of this carried out study?

5. How to select and optimize the user-defined parameters in the proposed model?

6. There are quite a few abbreviations are used in the manuscript. It is suggested to use a table to host all the frequently used abbreviations with their descriptions to improve the readability.

7. Some sentences are too long to follow, it is suggested that to break them down into short but meaningful ones to make the manuscript readable.

8. Explain the evaluation metrics and justify why those evaluation metrics are used?

9. It seems that the authors used images of equations, please use editable equation format.

10. The Related Works section is also fair, yet the criteria behind the selection of the works described should be explained.

11. The title is pretty deceptive and does not address the problem completely.

12. Every time a method/formula is used for something, it needs to be justified by either (a) prior work showing the superiority of this method, or (b) by your experiments showing its advantage over prior work methods - comparison is needed, or (c) formal proof of optimality. Please consider more prior works.

13. The data is not described. Proper data description should contain the number of data items, number of parameters, distribution analysis of parameters, and of the target parameter itself for classification.

14. The related works section is very short and no benefits from it. I suggest increasing the number of studies and add a new discussion there to show the advantage.

15. Method description is detailed and overall convincing, yet there is a big formatting problem with the end of this section and the beginning of the following one.

16. Figures 1-4 are low quality and very unprofessional figures please improve them.

17. Use Anova test to record the significant difference between performance of the proposed and existing methods.

6. PLOS authors have the option to publish the peer review history of their article (what does this mean?). If published, this will include your full peer review and any attached files.

Reviewer #1: No

Reviewer #2: No

---

## [Author Response · Author response to Decision Letter 0]

10 Aug 2021

Response to reviewers was supplied in the Files.

---

## [Decision Letter · Decision Letter 1]

24 Aug 2021

PONE-D-21-18748R1

easyFulcrum: An R package to process and analyze ecological sampling data generated using the Fulcrum mobile application

PLOS ONE

Dear Dr. Andersen,

Thank you for submitting your manuscript to PLOS ONE. After careful consideration, we feel that it has merit but does not fully meet PLOS ONE’s publication criteria as it currently stands. Therefore, we invite you to submit a revised version of the manuscript that addresses the points raised during the review process.

We look forward to receiving your revised manuscript.

Kind regards,

Syed Ahmad Chan Bukhari

Academic Editor

PLOS ONE

Journal Requirements:

Additional Editor Comments (if provided):

Reviewer 1 is still concerned. Would you mind revising accordingly?

Reviewers' comments:

Reviewer's Responses to Questions

**Comments to the Author**

1. If the authors have adequately addressed your comments raised in a previous round of review and you feel that this manuscript is now acceptable for publication, you may indicate that here to bypass the “Comments to the Author” section, enter your conflict of interest statement in the “Confidential to Editor” section, and submit your "Accept" recommendation.

Reviewer #1: All comments have been addressed

Reviewer #2: All comments have been addressed

2. Is the manuscript technically sound, and do the data support the conclusions?

Reviewer #1: Yes

Reviewer #2: Yes

3. Has the statistical analysis been performed appropriately and rigorously? 

Reviewer #1: N/A

Reviewer #2: N/A

4. Have the authors made all data underlying the findings in their manuscript fully available?

Reviewer #1: Yes

Reviewer #2: Yes

5. Is the manuscript presented in an intelligible fashion and written in standard English?

Reviewer #1: Yes

Reviewer #2: Yes

6. Review Comments to the Author

Reviewer #1: In this study authors are explaining how to use the R-package apps Fulcrum and easyFulcrum which have been used by them in previous study published in eLife (https://elifesciences.org/articles/50465). In this paper authors introduce easyFulcrum to combine Fulcrum data with genotyping data to make sampling easy.

1. easyFulcrum installation in the methods can be more elaborated to make it user friendly. Authors can simplify how to install R from devtools::install_github ("AndersenLab/easyfulcrum”), Imager and other prerequisites.

2. Genotyping templates have the option of entering Blast results but authors do not explain how the actual sequences of the PCR products linked to the C- and S- label in fulcrum.

3. Authors should mention examples of Molecular barcodes that can be used to identify organisms like DNA or RNA sequencing and lipid profiling etc.

Reviewer #2: The manuscript is relatively improved. Moreover, the manuscript is in a better format in terms of technicality and other results. I suggest accepting it.

7. PLOS authors have the option to publish the peer review history of their article (what does this mean?). If published, this will include your full peer review and any attached files.

Reviewer #1: No

Reviewer #2: No

---

## [Author Response · Author response to Decision Letter 1]

25 Aug 2021

Our responses to reviewer #1 are included as a separate PDF

---

## [Editor Report · Decision Letter 2]

22 Sep 2021

easyFulcrum: An R package to process and analyze ecological sampling data generated using the Fulcrum mobile application

PONE-D-21-18748R2

Dear Dr. Andersen,

We’re pleased to inform you that your manuscript has been judged scientifically suitable for publication and will be formally accepted for publication once it meets all outstanding technical requirements.

Kind regards,

Syed Ahmad Chan Bukhari

Academic Editor

PLOS ONE

---

## [Editor Report · Acceptance letter]

27 Sep 2021

PONE-D-21-18748R2 

easyFulcrum: An R package to process and analyze ecological sampling data generated using the Fulcrum mobile application 

Dear Dr. Andersen:

I'm pleased to inform you that your manuscript has been deemed suitable for publication in PLOS ONE. Congratulations! Your manuscript is now with our production department. 

Kind regards, 

on behalf of

Dr. Syed Ahmad Chan Bukhari 

Academic Editor

PLOS ONE